# Integrative Analysis of *GATA3* Expression and Variants as Prognostic Biomarkers in Urothelial Cancer

**DOI:** 10.3390/ijms26136378

**Published:** 2025-07-02

**Authors:** Chia-Min Chung, Han Chang, Chao-Hsiang Chang, Yi-Huei Chang, Po-Jen Hsiao, Chi-Shun Lien, Chi-Jung Chung

**Affiliations:** 1Graduate Institute of Biomedical Sciences, China Medical University, Taichung 404328, Taiwan; ibmsn141@gmail.com; 2Department of Psychiatry and Center for Addiction and Mental Health, China Medical University Hospital, Taichung 404327, Taiwan; 3Department of Pathology, College of Medicine, China Medical University Hospital, China Medical University, Taichung 404327, Taiwan; changhan2252@gmail.com; 4Department of Urology, College of Medicine, China Medical University Hospital, China Medical University, Taichung 404327, Taiwan; urology8395@yahoo.com.tw (C.-H.C.); yihueichang1006@gmail.com (Y.-H.C.); hspeter@gmail.com (P.-J.H.); r823918@gmail.com (C.-S.L.); 5Department of Public Health, College of Public Health, China Medical University, Taichung City 406040, Taiwan; 6Department of Medical Research, China Medical University Hospital, Taichung 404327, Taiwan

**Keywords:** GATA3 expression, urothelial carcinoma, survival, XGBoost-based machine learning, single-nucleotide polymorphism

## Abstract

*GATA3* is a transcription factor involved in urothelial differentiation and is widely used as a diagnostic marker for urothelial carcinoma (UC). Although loss of GATA3 expression has been linked to more aggressive disease, its prognostic significance remains uncertain. Genetic variation within the *GATA3* locus, particularly rs1244159, may influence protein expression and clinical outcomes. We conducted a case control study in Taiwan including 461 UC cases and 586 controls genotyped for four *GATA3* SNPs. GATA3 expression was assessed via immunohistochemistry (IHC) in 98 tumor tissues. Logistic regression and Kaplan–Meier analyses were used to evaluate SNP associations and survival outcomes. An XGBoost-based machine learning model with SHAP (SHapley Additive exPlanations) was applied to rank survival predictors. The rs1244159 G allele was associated with a significantly reduced UC risk (adjusted OR = 0.48, *p* = 0.0231) and higher GATA3 expression (*p* = 0.0173). High GATA3 expression predicted improved overall survival (*p* = 0.0092), particularly among G allele carriers (*p* = 0.0071). SHAP analysis identified age, chemotherapy, and GATA3 expression as the top predictors of survival, consistent with Cox regression results. In conclusion, our integrative analysis suggests that the rs1244159 G allele modulates GATA3 expression and influences UC prognosis. Combining genomics, pathology, and machine learning, *GATA3* may serve as a clinically useful biomarker for risk stratification and outcome prediction in UC.

## 1. Introduction

Urothelial carcinoma (UC) is the most common malignancy of the urinary tract, accounting for approximately 85% of bladder cancers. While non-invasive UC represents the majority of cases, about 25–30% present as muscle-invasive bladder cancer at the time of diagnosis, often associated with poorer prognosis and limited treatment options [1,2]. The substantial clinical and molecular heterogeneity of UC reflects underlying genetic and environmental complexity, posing significant challenges to risk stratification and the development of precision therapeutic strategies.

Genome-wide association studies (GWASs) have identified numerous genetic loci associated with UC susceptibility across diverse populations, offering valuable insights into its complex etiology [3,4]. However, most of these loci confer only modest individual effects, and their biological significance remains incompletely understood [5]. Although follow-up investigations have begun to elucidate the molecular mechanisms underlying some of these associations, functional validation of UC risk variants remains a major challenge [6,7]. Additionally, the development of accurate in vivo models for invasive bladder cancer is still lacking, underscoring the need for integrative approaches to link germline genetic variation with tumor biology and clinical outcomes [8,9].

Among these candidate genes, the transcription factor *GATA3*, a well-known lineage specific marker in breast and urothelial tissues, has garnered attention due to its dual role as a differentiation regulator and potential tumor suppressor [10]. *GATA3* is a zinc-finger transcription factor critical for urothelial cell differentiation and is widely utilized as an immunohistochemical marker for UC diagnosis [11,12]. While its loss is frequently observed in high-grade or muscle-invasive bladder cancers, the prognostic implications of GATA3 expression remain controversial, with conflicting findings reported in the literature [13,14]. Moreover, germline polymorphisms within the *GATA3* locus may contribute to inter-individual variation in gene expression, potentially influencing tumor biology and clinical outcomes.

To better understand the interaction between genetic susceptibility and GATA3 protein expression in UC, we conducted an integrative analysis combining tissue microarray-based immunohistochemistry (IHC) with SNP genotyping of candidate variants within the *GATA3* locus. We focused in particular on rs1244159, a putative regulatory SNP, to examine whether allelic variation is associated with differential GATA3 expression and clinical outcomes. By leveraging both conventional statistical models and machine learning-based interpretability, this study aims to clarify the prognostic significance of *GATA3* in UC and provide insights into how germline variation may inform personalized risk stratification and early intervention strategies.

## 2. Results

### 2.1. Participant Characteristics and Risk Factors

Descriptive analysis of demographic and clinical variables revealed no significant difference in age or sex distribution between UC patients and controls. However, UC cases were significantly more likely to have lower educational attainment (*p* < 0.0001), a history of cigarette smoking (*p* = 0.0001), and higher prevalence of comorbidities including hypertension (*p* = 0.0028), type 2 diabetes (*p* = 0.0021), chronic kidney disease (CKD) (*p* < 0.0001), and urinary stones (*p* = 0.0003), compared to controls. (Table 1).

### 2.2. Association of GATA3 SNPs with UC Risk

The association between *GATA3* polymorphisms and UC risk is presented in Table 2. Among the four SNPs analyzed, rs1244159 demonstrated a significant association with UC susceptibility. Compared to individuals with the A/A genotype, those homozygous for the G allele (G/G) had a significantly lower risk of UC (adjusted OR = 0.48, 95% CI: 0.27–0.87, *p* = 0.0231). An allele-based analysis further supported this finding, with the G allele associated with a reduced risk of UC (adjusted OR = 0.76, 95% CI: 0.61–0.95, *p* = 0.0171). No significant associations were found for rs12776126, rs11255526, or rs11255528 in either genotype- or allele-based models (Table 2).

### 2.3. *GATA3* Expression by rs1244159 Genotype

We further compared clinical features and GATA3 protein expression between UC patients carrying the A or G allele of rs1244159 (Table 3). G allele carriers were diagnosed at a significantly younger age than A allele carriers (mean age: 65.05 vs. 69.47 years, *p* = 0.0180). While no significant differences were observed in sex, smoking history, TNM stage, or treatment status between the two groups, G allele carriers exhibited a higher proportion of high GATA3 expression (50.98% vs. 33.10%, *p* = 0.0375). Although the mean H-score of GATA3 expression was higher among G allele carriers (96.82 vs. 85.41), the difference did not reach statistical significance (*p* = 0.0884).

### 2.4. Multivariate Survival Analysis and Machine Learning-Based Feature Importance

In multivariate Cox proportional hazards regression analysis, high GATA3 expression remained an independent favorable prognostic factor for overall survival after adjusting for clinicopathological covariates, with a hazard ratio of 0.37 (95% CI: 0.17–0.79, *p* = 0.0098) (Figure 1A). In contrast, the rs1244159 A/G allele showed no significant association with survival, and no significant interaction was observed between the rs1244159 genotype and GATA3 expression level (*p* = 0.1751).

To further explore the relative contribution of clinical and molecular variables, SHAP (SHapley Additive exPlanations) value analysis derived from an XGBoost model was applied. As shown in Figure 1B,C, age, chemotherapy, and GATA3 expression emerged as the top three predictors of survival, with mean SHAP values of 0.42, 0.32, and 0.24, respectively. These findings corroborate the results of traditional regression analysis and highlight the robust prognostic relevance of GATA3 protein expression in UC.

### 2.5. GATA3 Expression, rs1244159 Genotype, and UC Overall Survival

Kaplan–Meier survival analysis revealed that high GATA3 expression was significantly associated with improved overall survival in patients with UC (Figure 2A; log-rank *p* = 0.0177). Stratified analysis based on the rs1244159 genotype showed that this favorable prognostic effect was pronounced among carriers of the G allele (Figure 2B; log-rank *p* = 0.0294), but not among those with the A allele (Figure 2C; log-rank *p* = 0.1876).

When GATA3 expression was analyzed in combination with rs1244159 allele status, a trend toward better survival was observed in patients with high GATA3 expression, regardless of allele type; however, the difference did not reach statistical significance (Figure 2D; log-rank *p* = 0.0754). These results suggest a potential interaction between genetic variation and protein expression in modulating UC prognosis

## 3. Discussion

In this study, we investigated the clinical and biological significance of *GATA3* polymorphisms and protein expression in urothelial carcinoma (UC) through an integrative approach combining SNP genotyping and immunohistochemistry. Among the four *GATA3* variants examined, rs1244159 was significantly associated with UC risk, with the G allele linked to reduced susceptibility. G allele carriers also exhibited higher levels of GATA3 expression and were diagnosed at a younger age, suggesting a potential regulatory role of this variant. Although rs1244159 was not independently associated with survival, high GATA3 expression was consistently linked to favorable prognosis, particularly among carriers of the G allele, indicating a possible genotype dependent prognostic effect. To further evaluate the prognostic impact of clinical and molecular factors, SHAP-based feature importance analysis was conducted using XGBoost version 2.1.4, an open-source gradient boosting framework widely used for model interpretation. Age, chemotherapy, and GATA3 expression were identified as the top predictors of overall survival. These findings were concordant with results from multivariate Cox regression and support the relevance of *GATA3* as a robust prognostic marker. Overall, our results demonstrate the value of integrating genetic, pathological, and machine learning-based approaches to improve risk stratification and outcome prediction in UC.

The inverse association between GATA3 expression and the severity of UC has been documented in several studies, with loss of GATA3 expression more frequently observed in high-grade or muscle-invasive tumors [15,16]. Despite this, the prognostic value of GATA3 expression remains controversial. While five studies reported that GATA3-negative tumors were associated with poorer clinical outcomes compared to GATA3-positive tumors [10,12,17,18,19], some studies failed to detect any significant association [20,21,22], and one study even suggested a more favorable prognosis for GATA3-negative cases [23]. These conflicting findings highlight the complexity of GATA3’s role in UC progression and underscore the importance of considering additional molecular or genetic modifiers. Variability in tumor stage, histological subtype, and methodological differences in GATA3 detection may also contribute to the inconsistent results reported across studies.

In our study, high GATA3 expression was significantly associated with improved overall survival in UC patients, particularly among those carrying the rs1244159 G allele. The rs1244159 genotype correlated with GATA3 protein levels, suggesting that this variant may regulate gene expression and influence tumor behavior. G allele carriers exhibited both elevated GATA3 expression and an earlier age at diagnosis, implying a potential role in tumor initiation or progression.

Notably, rs1244159 was the only variant showing a significant association with UC risk, underscoring the possibility that *GATA3* regulation in UC is driven by a limited number of key functional variants rather than a broader polygenic model. This points to a more targeted genetic mechanism in urothelial tissue, with rs1244159 potentially acting as a regulatory SNP of functional relevance. While the precise molecular mechanism remains unclear, it is plausible that this variant affects transcription factor binding or chromatin accessibility, thereby modulating GATA3 expression.

By integrating genetic and protein expression data, our findings provide new insight into the molecular heterogeneity of UC. Identifying variants like rs1244159 that are linked to both expression and clinical outcomes can support more precise molecular stratification and inform tailored therapeutic strategies.

In addition to identifying genetic and molecular correlates, our study reaffirmed several established risk factors for UC, including cigarette smoking [24], lower educational attainment [25], and comorbid conditions such as CKD [26] and type 2 diabetes [27,28]. These findings underscore the multifactorial nature of UC and the complex interplay between genetic predisposition and environmental exposures in disease development. Importantly, the integration of *GATA3* variants [29] with clinical and lifestyle factors holds potential for refining risk stratification strategies and improving early detection efforts.

To address the limitations of a relatively modest sample size, we employed XGBoost, a machine learning algorithm known for its ability to handle small and heterogeneous datasets with high predictive accuracy. Rather than aiming to construct a complete mortality prediction model, we leveraged the strength of XGBoost to prioritize key prognostic features using SHAP (SHapley Additive ExPlanations) value analysis [30]. This approach revealed that age, chemotherapy, and GATA3 expression were the top predictors of survival, providing interpretable and clinically relevant insights. The consistency between SHAP-derived feature importance and multivariate Cox regression results further supports the robustness of our findings.

Collectively, our findings indicate that GATA3 expression, particularly when considered alongside the rs1244159 genotype, may serve as a valuable prognostic biomarker in UC. By integrating conventional statistical models with machine learning-based interpretability, this study highlights the power of combined analytic approaches to uncover clinically meaningful predictors, even in the context of limited sample sizes. Such strategies hold promise for advancing personalized risk stratification and guiding the development of targeted screening and intervention efforts in UC.

Despite the strengths of this study, several limitations should be acknowledged. First, the relatively small sample size in subgroup analyses, particularly those stratified by genotype and GATA3 expression, may have limited the statistical power to detect modest associations. To strengthen the robustness and generalizability of these findings, future studies should include larger, multi-center cohorts, incorporate validation datasets or meta-analyses, and explore the functional role of rs1244159 through experimental approaches to better elucidate its mechanistic impact on GATA3 expression and UC progression. Second, the observational nature of this study precludes definitive causal inferences. Third, as the study population was derived from a Taiwanese cohort, the generalizability of the findings to other ethnic groups may be limited due to potential population-specific genetic and environmental factors. Lastly, although rs1244159 emerged as the most significant variant in our analysis, its functional role remains to be elucidated. Further validation in independent cohorts and mechanistic studies are warranted to confirm its biological and clinical relevance.

## 4. Materials and Methods

### 4.1. Participant Recruitment, Questionnaires, and Biospecimen Collection

This study was based on a previously published medical center-based case–control study conducted between 2011 and 2019 in central Taiwan. A total of 687 patients aged 26–90 years with histologically confirmed UC—including 403 cases of bladder cancer and 284 cases of upper tract UC—were recruited at the Department of Urology, China Medical University Hospital. A control group of 1,514 individuals without a history of UC was randomly selected from participants undergoing routine health examinations at the Department of Family Medicine. The catchment area covered nearly the entire local population. For the present analysis, a total of 461 UC cases and 589 controls with available genotyping data were included.

Detailed demographic, lifestyle, and clinical data were collected at enrollment through structured, face-to-face interviews using standardized questionnaires and supplemented with medical record reviews. Information obtained included smoking status (never, former, or current smoker), education level, and family history of UC (yes/no). Cumulative cigarette smoking exposure was estimated by multiplying the duration of smoking (years) by the average number of packs smoked per day.

Clinically diagnosed comorbidities, hypertension, type 2 diabetes mellitus, urinary stones, and chronic kidney disease (CKD) were recorded as binary variables (yes/no). Renal function was further evaluated using estimated glomerular filtration rate (eGFR). Additionally, 5–6 mL of venous blood was collected to isolate leukocyte DNA for genotyping. This study was approved by the Institutional Review Board of China Medical University Hospital (CMUH107-REC1-147).

### 4.2. SNP Genotyping of GATA3 Locus

Genotype data for four single-nucleotide polymorphisms (SNPs) within the *GATA3* gene (rs11255526, rs12776126, rs1244159, and rs11255528) were extracted from the Axiom™ Genome-Wide Taiwan Biobank (TWB) array platform. Standard quality control (QC) procedures were applied to minimize potential biases arising from genotyping errors, sample contamination, or low-quality markers, which may lead to spurious associations [31].

QC was first performed at the individual level, where samples were excluded if they exhibited low genotyping call rates (<95%), gender discordance, or cryptic relatedness. Principal component analysis (PCA) was further applied to identify population outliers and adjust for potential population stratification.

Subsequently, QC was conducted at the marker level, excluding SNPs with a missing call rate >5%, minor allele frequency (MAF) <1%, or deviation from the Hardy–Weinberg equilibrium (HWE) in the control group (*p* ≤ 1 × 10^−5^). These stringent thresholds were implemented to ensure the robustness of downstream association analyses and to reduce both type I and type II errors.

### 4.3. Tissues Specimen Collection and Clinicopathological Features

From our initial cohort of 486 UC cases evaluated for *GATA3* SNP analysis, we randomly selected 98 cases with available tissue specimens for further investigation. Formalin-fixed, paraffin-embedded (FFPE) tissue specimens were retrieved from the archives of the Department of Pathology. Clinical parameters including age at diagnosis, sex, and smoking status were obtained from medical records and patient interviews. Pathological assessment included evaluation of tumor location, histological grade, morphological patterns, pathological stage, and molecular subtype classification. The presence of papillary architecture was documented based on microscopic examination. All specimens were reviewed and classified according to the 2016 World Health Organization (WHO) classification of urinary tract tumors. Pathological staging was performed according to the 8th edition of the American Joint Committee on Cancer (AJCC) TNM staging system, with tumors categorized as non-muscle invasive (Ta/Tis, T1) or muscle invasive (T2-T4), and further stratified by nodal (N0, N1-3) and metastatic (M0, M1) status. This subset of cases was subsequently analyzed for GATA3 expression levels and correlation with patient outcomes.

### 4.4. Tissue Microarray Construction and GATA3 Immunohistochemistry

Tissue microarrays (TMAs) were constructed using 98 UC specimens and 41 corresponding normal urothelial (NU) tissues obtained from transurethral resection, partial or radical cystectomy, or nephroureterectomy procedures. Tissue cores of 2 mm diameter were arrayed, and 4 μm sections were prepared for immunohistochemical analysis. TMA sections were deparaffinized in xylene, rehydrated through graded ethanol solutions, and equilibrated in phosphate-buffered saline. Antigen retrieval was performed in Tris-EDTA buffer (pH 9.0) using a pressure cooker (Cell Marque Corp., Hot Springs, AZ, USA) for 20 min. Immunohistochemical staining for GATA3 was performed using the BOND-MAX automated staining system (Leica Biosystems, Nussloch, Germany) with primary antibody against GATA3 (clone L50-823, 1:200 dilution; Biocare Medical, Pacheco, CA, USA). Nuclear immunoreactivity, visualized as brown chromogen deposition, was considered positive for GATA3 expression. Negative controls were prepared by substituting the primary antibody with normal mouse serum and phosphate-buffered saline (Figure 3). Quantitative analysis of GATA3 expression was conducted using the H-score method via digital image analysis software, Color Deconvolution v9 (Aperio Technologies Inc., Vista, CA, USA). The H-score (range: 0–300) was calculated based on both staining intensity and percentage of positively stained nuclei. The mean H-score of the 41 adjacent normal urothelial tissues was established as the threshold value for categorizing GATA3 expression in tumor samples as either high or low.

### 4.5. Statistical Analysis

All statistical analyses were conducted using SAS version 9.4 (SAS Institute, Cary, NC, USA). Descriptive statistics were used to summarize baseline demographic and clinical characteristics. Continuous variables were compared using Student’s *t*-tests, and categorical variables were analyzed using chi-square or Fisher’s exact tests, as appropriate. A two-tailed *p*-value < 0.05 was considered statistically significant.

To evaluate associations between *GATA3* single-nucleotide polymorphisms (SNPs) and UC risk, allele frequency analyses were conducted, followed by multivariable logistic regression models under additive and genotype models. Adjusted odds ratios (ORs) and 95% confidence intervals (CIs) were calculated, with adjustment for principal components (PC1–PC6) to account for population structure, age, sex, education level, cumulative smoking, type 2 diabetes, hypertension, CKD, and urinary stones.

For tissue-based analysis, GATA3 immunohistochemistry expression levels were compared between rs1244159 genotypes using chi-square tests for the categorical variable and Mann-Whitney U tests for the continuous variable. Kaplan–Meier survival curves were generated to assess the prognostic impact of GATA3 expression on overall survival. The log-rank test was used to compare survival distributions between high and low GATA3 expression groups, further stratified by the rs1244159 allele. To explore potential genotype-dependent effects, survival was further analyzed with stratification based on rs1244159 allele status (G or A allele) in combination with GATA3 expression levels (high or low), resulting in four distinct groups for comparison. Statistical significance was determined using the log-rank test with a threshold of *p* < 0.05.

To quantify and rank the contribution of multiple variables to survival outcomes, we implemented a machine learning approach based on extreme gradient boosting (XGBoost) algorithms with a Cox proportional hazards objective function, incorporating both survival time and event status (death) in the model. The XGBoost survival model was configured with a learning rate of 0.05 and maximum tree depth of 3, using Cox negative log-likelihood as the evaluation metric. After model development, SHapley Additive exPlanations (SHAP) values were calculated to estimate the contribution of each variable, based on cooperative game theory providing both global importance measurements and individual-level impact assessments [32]. The absolute SHAP values were averaged across all patients to calculate mean absolute SHAP values (mean |SHAP|), which were then used to rank variables according to their relative importance in predicting survival. Additionally, SHAP value distributions were visualized to illustrate the direction and magnitude of each variable’s impact on survival probability. When assessed on the same dataset, the XGBoost survival model yielded a concordance index (C-index) of 0.940, compared to 0.686 for the traditional Cox proportional hazards model. All analyses were performed using the Python implementation of XGBoost (version 1.5.1) and SHAP (version 0.40.0) packages.

## 5. Conclusions

Our integrative analysis reveals that the *GATA3* rs1244159 polymorphism plays a modulatory role in UC, influencing both GATA3 protein expression and disease susceptibility. While the G allele was associated with reduced UC risk and elevated GATA3 expression, high GATA3 levels independently predicted better overall survival, particularly among G allele carriers. Although rs1244159 itself was not directly linked to prognosis, its interaction with GATA3 expression highlights the importance of considering germline tumor biomarker relationships. By combining genomic, pathological, and machine learning-based approaches, this study demonstrates the potential clinical utility of GATA3 as a prognostic biomarker and supports its relevance in personalized risk stratification and outcome prediction in UC. However, these findings should be interpreted with caution due to limitations in the sample size and representativeness of the patient cohort, which may affect the generalizability and statistical confidence of the results. Further validation in larger, independent cohorts is warranted.

## Figures and Tables

**Figure 1 ijms-26-06378-f001:**
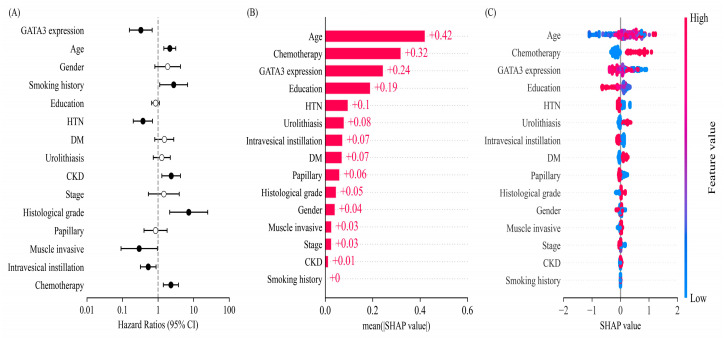
Multivariate Cox regression and SHAP-based feature importance for overall survival in UC patients. (**A**) Multivariate Cox proportional hazards regression analysis evaluating the effect of clinicopathological, genetic, and molecular features on overall survival in UC patients. The filled (black) circles indicate statistically significant results, while the open (white) circles indicate non-significant results. (**B**) Mean absolute SHAP values derived from an XGBoost model, indicating the relative importance of each variable in predicting overall survival. (**C**) SHAP summary plot showing the directional impact of each feature on survival probability. Red and blue dots represent high and low feature values, respectively.

**Figure 2 ijms-26-06378-f002:**
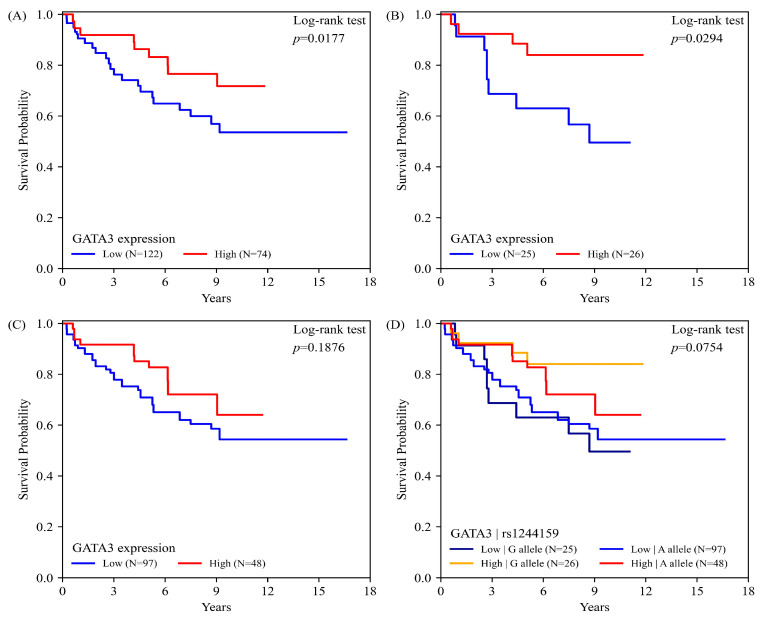
Kaplan–Meier curves depicting overall survival in urothelial carcinoma (UC) patients stratified by the GATA3 expression level and rs1244159 genotype. (**A**) Patients with high GATA3 expression exhibited significantly improved survival compared to those with low expression (log-rank *p* = 0.0177; Low: n = 122, High: n = 74). (**B**) Among rs1244159 G allele carriers, high GATA3 expression was associated with significantly better survival (log-rank *p* = 0.0294; Low: n = 25, High: n = 26). (**C**) Among A allele carriers, the difference in survival between high and low GATA3 expression groups was not significant (log-rank *p* = 0.1876; Low: n = 97, High: n = 48). (**D**) Joint stratification by GATA3 protein expression and the rs1244159 genotype was used to assess their potential interaction effect on survival. Patients with high GATA3 expression regardless of allele tended to show better outcomes, although this trend did not reach statistical significance (log-rank *p* = 0.0754; Low/A allele: n = 97, High/A allele: n = 48, Low/G allele: n = 25, High/G allele: n = 26).

**Figure 3 ijms-26-06378-f003:**
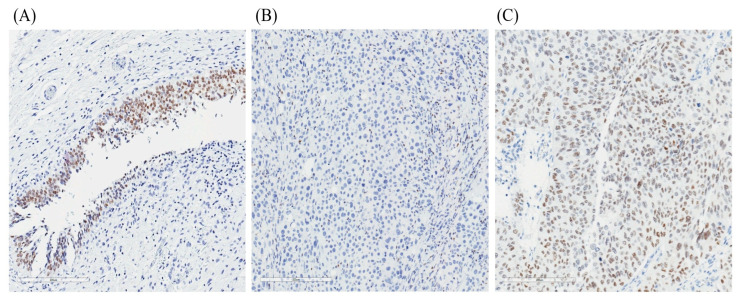
Representative images of GATA3 immunohistochemical staining in urothelial tissue. (**A**) Normal urothelium displaying diffuse nuclear GATA3 expression, serving as an internal positive control. (**B**) Urothelial carcinoma lacking GATA3 expression (GATA3-negative), characterized by an absence of nuclear staining in tumor cells. (**C**) Urothelial carcinoma (UC) with high GATA3 expression showing strong nuclear staining in tumor cells (GATA3-positive). All images were obtained from tissue microarrays using standard IHC protocols with anti-GATA3 antibody (clone L50-823). Scale bars = 200 μm.

**Table 1 ijms-26-06378-t001:** Descriptive characteristics, risk factors, and comorbidities in the study participants with urothelial carcinoma (UC) and without UC.

Variables	Control (*n* = 589)	UC Case (*n* = 461)	*p*-Value
Age; Mean (SD)	66.70 (11.08)	66.19 (11.51)	0.4669
age < 65	255 (43.29)	199 (43.26)	0.9672
Age ≥ 65	334 (56.71)	262 (56.74)	
Sex			
Female	256 (43.46)	192 (41.65)	0.5753
Male	333 (56.54)	269 (58.35)	
Education			
Elementary school or lower	173 (29.62)	220 (47.93)	<0.0001
High school	243 (41.61)	160 (34.64)	
College or higher	168 (28.77)	80 (17.43)	
Smoking history			
No	432 (73.34)	291 (62.4)	0.0001
Yes	157 (26.66)	170 (37.6)	
HTN			
No	368 (62.91)	241 (53.67)	0.0028
Yes	217 (37.09)	208 (46.33)	
DM			
No	513 (87.4)	361 (80.04)	0.0021
Yes	76 (12.6)	90 (19.96)	
CKD			
No	508 (86.25)	213 (46.20)	<0.0001
Yes	81 (13.75)	248 (53.80)	
Urinary stones			
No	497 (84.81)	342 (75.83)	0.0003
Yes	89 (15.19)	109 (24.17)	

HTN: hypertension; DM: diabetes mellitus; CKD: chronic kidney diseases.

**Table 2 ijms-26-06378-t002:** Associations between candidate SNPs in *GATA3* and UC with adjustment for risk factors and comorbidities.

SNP	Genotype	Control (%)	UC (%)	OR	*p* ^a^	Adjusted OR	*p* ^b^
rs12776126							
	G/G	460 (78.10)	364 (78.96)	Ref			
	G/T	122 (20.71)	96 (20.82)	0.97 (0.72–1.32)	0.1298	1.08 (0.76–1.52)	0.5728
	T/T	7 (1.19)	1 (0.22)	5.07 (0.62–41.70)	0.1281	2.18 (0.26–18.05)	0.4917
	T	136 (11.54)	98 (10.63)	1.12 (0.83–1.53)	0.4578	1.18 (0.86–1.63)	0.3146
	G ^c^	1042 (88.46)	824 (89.37)	Ref allele			
rs1244159							
	A/A	321 (54.50)	233 (50.65)	Ref			
	A/G	238 (40.41)	194 (42.17)	0.91 (0.70–1.17)	0.3502	0.88 (0.66–1.18)	0.2025
	G/G	30 (5.09)	33 (7.17)	0.60 (0.35–1.02)	0.0836	0.48 (0.27–0.87)	0.0231
	G	297 (25.26)	260 (28.26)	0.80 (0.64–0.99)	0.0373	0.76 (0.61–0.95)	0.0171
	A ^c^	879 (74.74)	660 (71.74)	Ref allele			
rs11255526							
	G/G	242 (41.30)	172 (37.39)	Ref			
	G/C	274 (46.76)	223 (48.48)	0.92 (0.70–1.20)	0.8366	0.95 (0.70–1.29)	0.9084
	C/C	70 (11.95)	65 (14.13)	0.80 (0.54–1.18)	0.3236	0.87 (0.56–1.36)	0.5994
	C	413 (35.30)	353 (38.37)	0.98 (0.80–1.20)	0.8599	0.94 (0.76–1.15)	0.5488
	G ^c^	757 (64.70)	567 (61.63)	Ref allele			
rs11255528							
	C/C	304 (51.61)	248 (54.03)	Ref			
	C/T	231 (39.22)	170 (37.04)	1.11 (0.85–1.44)	0.6107	1.07 (0.79–1.44)	0.8448
	T/T	54 (9.17)	41 (8.93)	1.05 (0.67–1.64)	0.9920	1.06 (0.64–1.76)	0.9118
	T	339 (28.83)	252 (27.45)	1.01 (0.81–1.24)	0.9903	1.05 (0.84–1.31)	0.6652
	C^c^	837 (71.17)	666 (72.55)	Ref allele			

^a^ Models were adjusted for principal components 1 to 6 (PC1–PC6) to account for population stratification. ^b^ Fully adjusted models included PC1–PC6, age, sex, cumulative cigarette smoking, hypertension, type 2 diabetes, chronic kidney disease (CKD), and urinary stones. ^c^ Logistic regression models for allele-based analyses were conducted using the major allele as the reference.

**Table 3 ijms-26-06378-t003:** Comparison of clinical and GATA3 protein expression in tissue samples between UC patients carrying A or G allele of rs1244159.

	A Allele (*n* = 145)	G Allele (*n*= 51)	*p* Value
Diagnosis age	69.47 ± 12.37	65.05 ± 11.81	0.0180
Gender			0.9014
Male	64 (44.14)	22 (43.14)	
Female	81 (55.86)	29 (56.86)	
Smoking history	54 (37.24)	14 (27.45)	0.2064
HTN	75 (51.72)	27 (55.10)	0.6823
DM	45 (31.03)	19 (38.78)	0.3191
CKD	89 (61.38)	31 (63.27)	0.8142
Urolithiasis	33 (22.76)	9 (18.37)	0.5188
TNM stage			0.1462
0a to I	74 (51.03)	20 (39.22)	
II to IV	71 (48.97)	31 (60.78)	
Histological grade			0.3459
Low grade	38 (26.21)	10 (19.61)	
High grade	107 (73.79)	41 (80.39)	
Papillary			0.4842
No	51 (35.66)	21 (41.18)	
Yes	92 (64.34)	30 (58.82)	
Muscle invasive			0.4233
No	45 (31.47)	13 (25.49)	
Yes	98 (68.53)	38 (74.51)	
Intravesical instillation			0.0789
No	89 (63.12)	25 (49.02)	
Yes	52 (36.88)	26 (50.98)	
Chemotherapy			0.5983
No	91 (64.54)	35 (68.63)	
Yes	50 (35.46)	16 (31.37)	
GATA3 expression	85. 41±45.17	96.82±49.61	0.0884
Low	97 (66.90)	25 (49.02)	0.0375
High	48 (33.10)	26 (50.98)	

HTN: hypertension; DM: diabetes mellitus; CKD: chronic kidney diseases.

## Data Availability

Due to the sensitive nature of the patient data collected for this study and privacy regulations, the raw datasets are not publicly available. A limited, de-identified dataset might be made available by the corresponding author upon reasonable request, pending approval from the institutional ethics committee.

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
