# Peer review of "Integrative Analysis of GATA3 Expression and Variants as Prognostic Biomarkers in Urothelial Cancer"

_ijms, 2025, doi:10.3390/ijms26136378_

Round 1
Reviewer 1 Report
Comments and Suggestions for Authors
The authors present a manuscript investigating the role of how genetic variation in the GATA3 gene, particularly the rs1244159 SNP, may influence both risk and prognosis in urothelial carcinoma. The authors combine data from a large Taiwanese case-control cohort with tumor-level protein expression analysis and apply both traditional statistical methods and machine learning to explore the clinical significance of GATA3.
This reviewer finds the manuscript to be well structured and well-designed, with the authors make good use of multiple analytical approaches. One limitation I find is that this work is based solely on a single Taiwanese population, and as the authors mention it limits it generalizability to other populations. Future studies should therefore be aimed at validating these findings in other populations where urothelial cancer is prevalent. However, this does not prevent publication and I recommend this manuscript be accepted.
Author Response
Response to Reviewer 1 Comments
We sincerely thank the reviewer for the positive evaluation of our manuscript and the constructive comments. Below, we address the reviewer’s points in detail.
Comment 1:
“The authors present a manuscript investigating the role of how genetic variation in the GATA3 gene, particularly the rs1244159 SNP, may influence both risk and prognosis in urothelial carcinoma. The authors combine data from a large Taiwanese case-control cohort with tumor-level protein expression analysis and apply both traditional statistical methods and machine learning to explore the clinical significance of GATA3.”
Response:
We thank the reviewer for recognizing the strength of our integrative approach combining genetic epidemiology, tissue-based pathology, and machine learning techniques to explore the prognostic significance of GATA3 in urothelial carcinoma.
Comment 2:
“One limitation I find is that this work is based solely on a single Taiwanese population, and as the authors mention it limits its generalizability to other populations. Future studies should therefore be aimed at validating these findings in other populations where urothelial cancer is prevalent.”
Response:
We agree with the reviewer’s observation. As recommended, we have emphasized this point more clearly in the revised Discussion section by noting the importance of validating our findings in independent cohorts across diverse ethnic backgrounds. We have also highlighted the need for collaborative international studies to further assess the broader clinical relevance of GATA3 genetic variants.
Final Recommendation:
“However, this does not prevent publication and I recommend this manuscript be accepted.”
Response:
We are grateful for the reviewer’s recommendation for acceptance and the thoughtful consideration of our work. We have implemented minor revisions as outlined above to further strengthen the manuscript.
Reviewer 2 Report
Comments and Suggestions for Authors
This manuscript reports an analysis of the expression of GATA3 as a potential prognostic biomarker in the development of urothelial cancer. The manuscript is mostly clearly written and appropriate supporting tables and figures are provided. Some limited insights into the value of GATA3 as a prognostic factor are included in this paper and appropriate statistical analyses have been completed. The paper modestly enhances our understanding of potential prognostic biomarkers in urothelial cancer. There are some points that I feel the authors should address and these are detailed below (all require changes to the manuscript).
- Figure 2. Can you include the relevant sample size (n), i.e. the number of patients for each comparison plot so that the figure is comprehensible independent of the rest of the manuscript and so the reader can quickly and easily see what impact sample size might have on the results.
- Page 8 lines 206-207 “The inverse association . . . or muscle invasive tumors.” Both these two statements need supporting references.
- Page 8 lines 224-226 “These findings underscore . . . stratification in UC.” I think this statement is rather vague and circuitous. Can you expand in more detail on exactly how your results facilitate molecular stratification in urothelial cancer? Also, I think you should expand more on how studies of the genetic variation should be coordinated with detailed examination of the protein expression.
- In this paper, you only found one variant (rs1244159) that showed significant association with risk of urothelial cancer. You haven’t discussed this fact much in the paper. What does it reveal about the genetics associated with GATA3 and corresponding impacts on urothelial cancer risk?
- Page 9 lines 250-253 “First, the sample size . . . modest associations.” This reinforces point 1 above. Also, what practical next steps could be taken to improve the statistical confidence and resolution of the comparisons?
- Figure 3 legend. The figure legend should be given immediately below the corresponding figure and not with main text inserted in between.
- Page 12 lines 391-394 “By combining genomic . . . prediction in UC.” This statement is of course subject to caveats regarding the size and representativeness of the patient sample set in this study.
The manuscript needs an editorial check to correct minor typographical errors.
Author Response
Response to Reviewer #2 Comments
We thank the reviewer for the thoughtful and constructive feedback. Below we provide point-by-point responses to each comment, and corresponding revisions have been made in the manuscript where appropriate.
Comment 1: Figure 2. Can you include the relevant sample size (n), i.e., the number of patients for each comparison plot so that the figure is comprehensible independent of the rest of the manuscript and so the reader can quickly and easily see what impact sample size might have on the results.
Response: We agree with the reviewer’s suggestion. The sample sizes (n) for each Kaplan–Meier subgroup plot have been added to the figure panels and are now also indicated in the revised Figure 2 legend. This allows for easier interpretation of the results and better understanding of the impact of sample size on the survival comparisons.
Comment 2: Page 8, lines 206–207: “The inverse association . . . or muscle invasive tumors.” Both these two statements need supporting references.
Response:
We thank the reviewer for noting this. We have now added relevant references to support the association between GATA3 loss and more aggressive urothelial carcinoma, including studies that reported its frequent downregulation in high-grade and muscle-invasive tumors. Refs added as follows:
- Miyamoto H, Izumi K, Yao JL, et al. GATA binding protein 3 is down-regulated in bladder cancer yet strong expression is an independent predictor of poor prognosis in invasive tumor. Human pathology. 2012;43(11):2033-40. doi:10.1016/j.humpath.2012.02.011
- Wang CC, Tsai YC, Jeng YM. Biological significance of GATA3, cytokeratin 20, cytokeratin 5/6 and p53 expression in muscle-invasive bladder cancer. PloS one. 2019;14(8):e0221785. doi:10.1371/journal.pone.0221785
Comment 3: Page 8, lines 224–226: “These findings underscore . . . stratification in UC.” This statement is rather vague and circuitous. Can you expand in more detail on exactly how your results facilitate molecular stratification in urothelial cancer? Also, I think you should expand more on how studies of the genetic variation should be coordinated with detailed examination of the protein expression.
Response:
Thank you for this valuable suggestion. We have revised the text to clearly explain how our results facilitate molecular stratification in urothelial carcinoma (UC). Specifically, we observed that high GATA3 expression was significantly associated with improved overall survival, particularly among patients carrying the rs1244159 G allele. Our findings demonstrate a correlation between rs1244159 genotype and GATA3 protein levels, suggesting that this genetic variant likely regulates GATA3 expression, potentially influencing tumor initiation or progression. We hypothesize that rs1244159, possibly located near a regulatory region, might modulate transcription factor binding or chromatin accessibility, thereby affecting gene expression. (Page 8, lines 224–230).
Comment 4: In this paper, you only found one variant (rs1244159) that showed significant association with risk of urothelial cancer. You haven’t discussed this fact much in the paper. What does it reveal about the genetics associated with GATA3 and corresponding impacts on urothelial cancer risk?
Response:
Thank you for highlighting this. We have now included a paragraph in the Discussion section to address this point. Specifically, we discuss the implication that rs1244159 may reside in or be linked to a regulatory region affecting GATA3 expression, and that the lack of association for other variants may reflect either limited functional impact or insufficient power to detect small effects. We also note that this supports a more targeted approach for functional follow-up studies focused on rs1244159(Page 8, lines 218–223).
Comment 5: Page 9, lines 250–253: “First, the sample size . . . modest associations.” This reinforces point 1 above. Also, what practical next steps could be taken to improve the statistical confidence and resolution of the comparisons?
Authors' Response:
Thank you for pointing this out. We acknowledge that the sample size, particularly in subgroup analyses stratified by genotype and GATA3 expression, may limit the statistical power to detect modest associations. To address this limitation practically, future studies could expand patient recruitment to increase sample sizes and conduct multi-center collaborations to enhance the representativeness and generalizability of the findings. Additionally, incorporating validation cohorts or meta-analyses with external datasets would further improve statistical confidence and refine the resolution of observed associations. Experimental validation of the functional impact of the rs1244159 variant on GATA3 expression and tumor biology could also provide deeper mechanistic insights and support more definitive causal interpretations. These considerations have been reflected in the revised limitations section (Page 9, lines 259–267).
Comment 6: Figure 3 legend. The figure legend should be given immediately below the corresponding figure and not with main text inserted in between.
Response:
Thank you for pointing this out. We have corrected the figure layout in the revised manuscript to ensure that the legend appears directly below Figure 3, as per standard formatting guidelines.
Comment 7: Page 12, lines 391–394: “By combining genomic . . . prediction in UC.” This statement is of course subject to caveats regarding the size and representativeness of the patient sample set in this study.
Authors' Response:
Thank you for this important comment. We agree that the interpretation of our findings must be considered within the context of the study's sample size and cohort characteristics. Accordingly, we have revised the sentence to acknowledge this limitation (Page 12, lines 408–414)